# Skill-Based Mixture-of-Experts:
# Adaptive Routing for Heterogeneous Reasoning via Inferred Skills

**Justin Chih-Yao Chen** [* 1]  **Sukwon Yun** [* 1]  **Elias Stengel-Eskin** [* 2]  **Tianlong Chen** [1]  **Mohit Bansal** [1]

## Abstract

Combining existing pre-trained LLMs is a promising approach for diverse reasoning tasks. However, task-level expert selection is often too coarse-grained, since different instances may require different expertise. To address this, we propose SKILL-MOE, a symbolic, skill-based, and gradient-free Mixture-of-Experts framework for instance-level expert selection. SKILL-MOE infers skills (e.g., *algebra* in mathematics) from each query, selects experts based on skill relevance, and lets each expert generate its own reasoning. The resulting $k$ outputs are then synthesized by an aggregator chosen for its ability to integrate diverse responses. While instance-level selection substantially improves performance, naively implementing it incurs heavy overhead from repeated model loading and offloading. We address this with a batch inference strategy that groups instances by assigned experts, allowing each model to be loaded only once. As a result, SKILL-MOE integrates 16 expert models *on a single GPU* with runtime comparable to prior multi-agent baselines using 4 GPUs. Across diverse benchmarks (MMLU-Pro, GPQA, AIME, and MedMCQA), SKILL-MOE achieves an average absolute improvement of $8.15\%$ over the best baseline. It also generalizes well to unseen tasks and outperforms discussion-based methods without requiring expensive multi-round interactions.[1]

## 1. Introduction

A core strength of humans is our ability to communicate and coordinate with each other using language (Clark, 1996; Yow & Lim, 2019; Xu et al., 2023). This allows diverse experts to contribute specialized knowledge towards solving a problem. Like humans, large language models (LLMs) often have differing skills and strengths, derived from differences in their architectures and training regimens. For instance, math-specific models like MetaMath (Yu et al., 2023) or QwenMath (Yang et al., 2024) are post-trained with mathematical reasoning data, making them particularly adept at math tasks – often at the cost of performance on out-of-distribution tasks (Kumar et al., 2022; Chu et al., 2025) like commonsense or medical reasoning (Lobo et al., 2024). Even within specialized domains, differences in pre-training data can lead to nuanced variations in expertise: one math-focused model may excel at algebra, while another is better suited for geometry. This motivates our development of a skill-based framework designed to identify and select the most suitable set of expert models *for each problem*. Indeed, combining multiple "expert" models via Mixture-of-Experts (MoE) is well-studied (Jacobs et al., 1991; Eigen et al., 2013) and has been applied widely for large pre-trained models, enabling better performance at a lower computational cost (Shazeer et al., 2017a; Fedus et al., 2022; Riquelme et al., 2021). However, in the conventional MoE settings, experts are typically sub-models, i.e., subsets of parameters within a larger model, where at test time, they are combined in the model's parameter space. This generally requires end-to-end training from scratch, which is often computationally expensive and precludes the re-use of the vast pool of already-trained LLMs. Building on recent efforts in combining a fixed set of models through multi-agent discussions (Chen et al., 2024b; Du et al., 2023; Liang et al., 2023; Wang et al., 2024a), we propose exploring a new *training-free* paradigm for large-scale MoEs: a skill-based mixture of experts (SKILL-MOE). Rather than using information encoded in the model's hidden state, SKILL-MOE uses symbolic structures in two ways: First, SKILL-MOE infers a set of discrete skills needed to solve a problem, measuring the abilities of each model in a pool of candidate expert models. It then uses skill-based performance as a "router" to recruit a sparse subset of experts *for each problem*. Secondly, SKILL-MOE combines pre-trained experts through a symbolic channel, i.e., language, which is a common protocol already shared by all LLMs. To take advantage of the diverse set of expert LLMs, we must address

---

[*]Equal contribution [1]UNC Chapel Hill [2]The University of Texas at Austin. Correspondence to: <cychen@cs.unc.edu>.

*Proceedings of the 43rd International Conference on Machine Learning*, Seoul, South Korea. PMLR 306, 2026. Copyright 2026 by the author(s).

[1]Code: https://github.com/dinobby/Skill-MoE

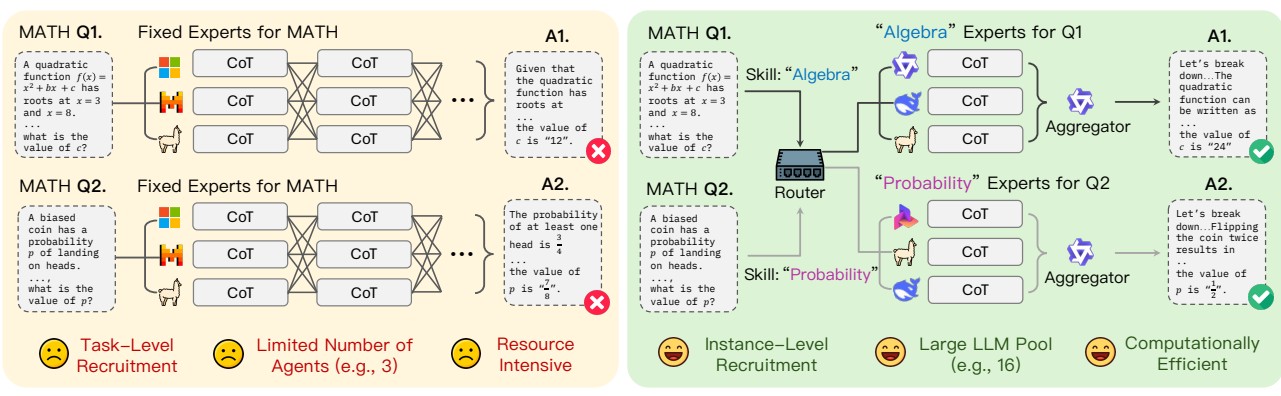

*Figure 1.* (a) In prior work, a fixed set of task-level experts is recruited to solve mathematical problems, while heterogeneous questions may differ in the skills required to solve them (e.g., Q1 requires algebra, while Q2 focuses on probability). The recruited experts generate outputs for multiple rounds, making these methods inefficient. (b) In contrast, SKILL-MOE adaptively recruits instance-level experts based on skills needed ("Algebra" experts for Q1 and a different set of "Probability" experts for Q2). By generating only a single round of responses with an aggregator to synthesize the final output, our approach is both more performant and more efficient.

two key challenges: **(1) Effective Expert Selection**: Given a large set of LLMs, how can we choose the best experts for each instance? **(2) Scalable Expert Mixing**: How can we serve a large number of experts (e.g. 16) without increasing the demand for GPUs?

**(1) Effective Expert Selection:** The increasing diversity of benchmarks (Miranda et al., 2024) and the growing number of models means that experts must be selected not at the level of tasks, but at the level of individual queries. Even at the task level, manual selection can be labor-intensive, and the performance of multi-agent frameworks can be sensitive to the agents chosen for a task (Chen et al., 2024b; Liang et al., 2023; Wang et al., 2024b). For instance, as shown in Figure 1 (a), while a given subset of models may perform well on math tasks on average, their proficiency in specific subfields like algebra or probability might vary – that is, using a fixed subset of models on all math samples might hurt performance on particular subtasks. This underscores the need for a finer-grained selection mechanism, as shown in Figure 1 (b).

**(2) Scalable Expert Mixing:** Past work has often relied on multiple rounds of inference, leading to significant GPU demands. Moreover, it does not scale to a dynamic setting like the one we consider, where the number of GPUs required would be equal to the number of potential models available (in our case, 16), making this option prohibitively expensive. We instead propose a **batch inference mechanism** that groups samples into batches per expert. We then run *all queries* assigned to a given expert at once, reducing the time for constant loading and offloading models, which is far faster than sequential processing. While this strategy accommodates 16 models *on a single GPU*, it can also work with multiple GPUs for improved speed.

We evaluate SKILL-MOE on diverse benchmarks across math, science, and medical reasoning, using a diverse model pool consisting of 16 experts. We show that our automated skill-based recruiting yields an average accuracy improvement of 8.15% over the best multi-agent baseline. Moreover, SKILL-MOE consistently surpasses all baselines, whereas the strongest baseline changes across tasks. Therefore, our method eliminates the need to evaluate and compare a large number of baselines for each task. Notably, using a single GPU, SKILL-MOE has 44% less inference run-time than a mixture-of-agents baseline (Wang et al., 2024a); when four GPUs are available for both methods, we obtain an almost 2× speedup. Finally, our analysis shows that SKILL-MOE generalizes well to unseen tasks (up to +14.81% compared to the strongest baseline), and selecting a task-specific aggregator achieves performance comparable to multi-round discussion while requiring substantially less compute.

## 2. Related Work

**Mixture-of-Agents.** Traditional Mixture-of-Experts (MoE) models (Jacobs et al., 1991; Jordan & Jacobs, 1994; Chen et al., 1999; Yuksel et al., 2012) distribute computation across multiple "expert" submodels. The Sparse MoE (SMoE) approach (Shazeer et al., 2017a) improves efficiency by activating only the most relevant experts per input. This method has been widely applied in vision tasks (Riquelme et al., 2021; Wang et al., 2020; Yang et al., 2019; Abbas & Andreopoulos, 2020), language tasks (Lepikhin et al., 2021; Zhang et al., 2022; Zuo et al., 2022; Jiang et al., 2021) and multimodal learning (Kudugunta et al., 2021; Yun et al., 2024). Unlike SKILL-MOE, standard MoE approaches require experts to be trained jointly, with communication taking place in the parameter spaces

of the submodels. LLM routing has also been an active research area. DER formulates routing as a Markov Decision Process and dynamically selects an optimal answering route (Hu et al., 2025). Router-R1 formulates routing and aggregation as a sequential decision process (Zhang et al., 2025). On the test-time mixture side, LLM-Blender proposes ranking and fusing multiple models' output (Jiang et al., 2023). MoA (Wang et al., 2024a) combine LLM agents into ensembles that rely on a fixed set of agents. This approach requires multiple rounds of generation and aggregation before producing a final answer. Similarly, Self-MoA (Li et al., 2025) suggest that optimal performance can be achieved by invoking the task-best model multiple times alongside the task-best aggregator. Our work differs from past routing work and test-time mixture like MoA and Self-MoA by introducing *adaptive, instance-level, skill-based routing* while avoiding costly multi-model discussions in favor of streamlined aggregation.

**Multi-Agent Reasoning.** Multi-agent reasoning has emerged as a promising paradigm for enhancing complex problem-solving and decision-making in AI systems. Early approaches employed reinforcement learning-based coordination (Lowe et al., 2017; Foerster et al., 2018; Jaques et al., 2019), while recent efforts leverage LLM-based multi-agent frameworks. One line of research explores student-teacher paradigms (Magister et al., 2022; Fu et al., 2023; Ho et al., 2022; Du et al., 2023; Chen et al., 2024a), where reasoning capabilities are distilled from stronger to weaker agents. Another approach investigates multi-agent debate frameworks, where agents interact to refine arguments and enhance collective decision-making; this has been explored with multiple instances of a single model (Liang et al., 2023; Xiong et al., 2023; Chan et al., 2023) or debates between multiple LLM types (Chen et al., 2024b). In both cases, the set of models is predefined by the user. In contrast, our approach automatically selects models based on inferred skills. Additionally, our framework achieves superior efficiency by avoiding multi-round discussions.

## 3. Methodology

### 3.1. Problem Setup

Given a pool of $n$ models $\mathcal{M} = \{M_i\}_{i=1}^n$, where each model represents a distinct LLM with potentially different pre-training datasets and architectures, our goal is to optimize performance through dynamic allocation – solving each problem with the most suitable subset of $k$ models, allowing experts to combine information to enhance reasoning. To achieve this, we assume access to a small validation set to obtain (1) model profiles $P_i \, \forall i \leq n$, and (2) aggregator profiles that benchmark the ability of each model to act as an aggregator. We use these profiles to recruit experts (at the instance level) and to select the aggregator (at the task

level). Below, we describe how these profiles are created.

### 3.2. Preprocessing

#### 3.2.1. MODEL PROFILE CREATION

To recruit the $k$ most suitable experts for a given question, we assess each model's specialized skills across various problem-solving domains, illustrated in Figure 2 (a). This is done by evaluating their performance on the validation set for each task (see Table 15 for sizes), thereby constructing a model profile $P_i$ for each model $M_i$. For each question in the validation set, we first prompt an LLM – referred to as the "Keyword LLM" – to identify the essential skills required to solve the problem. For consistency, we use Qwen2.5-7B-Instruct (Qwen Team, 2024) as the Keyword LLM. Later in Table 16, we show that the choice of the keyword LLM has little effect on the performance. To reduce noise, we generate keyword annotations for each question five times, and retain only those that appear more than once for each question. These extracted skills represent core knowledge areas necessary for solving the problem – for instance, a given college-level math problem may require skills such as algebra, calculus, and geometry. Once all questions are annotated with their required skills, each model $M_i$ in the pool attempts to solve them using Chain-of-Thought reasoning (Wei et al., 2022). A correct answer increases the score of each associated skill by $+1$, while an incorrect answer results in a $-1$ penalty. At the end of this process, each model has a profile $P_i$ represented as a dictionary, e.g., {'Algebra': 10, 'Biology': 3, 'Chemistry': -6, ...}.

#### 3.2.2. AGGREGATOR SELECTION

An aggregator is a model that consolidates $k$ outputs into a single high-quality response. Our pilot experiments, along with findings from Wang et al. (2024a) and Li et al. (2025), indicate that the aggregator model plays a crucial role in the final performance, and selecting the most effective model for aggregation is non-trivial. We find that *a strong reasoning model is not necessarily a strong aggregator and vice versa*; qualitatively, we show this later in Table 3. We also find that adaptively selecting an aggregator on the instance level based on model profiles is less effective, motivating us to choose the aggregator based on its ability to select correct answers. To identify the best aggregator per task, we construct a synthetic task using the same validation set. From the profile creation process, we obtain outputs from all models, some correct and some incorrect. For each question, we sample one correct reasoning chain and two incorrect ones, structuring the input as follows: {question}, {correct_CoT}, {incorrect_CoT}, {incorrect_CoT}. We shuffle the order of the CoTs and instruct each model to act as an aggregator (using the prompt shown in Appendix R), syn-

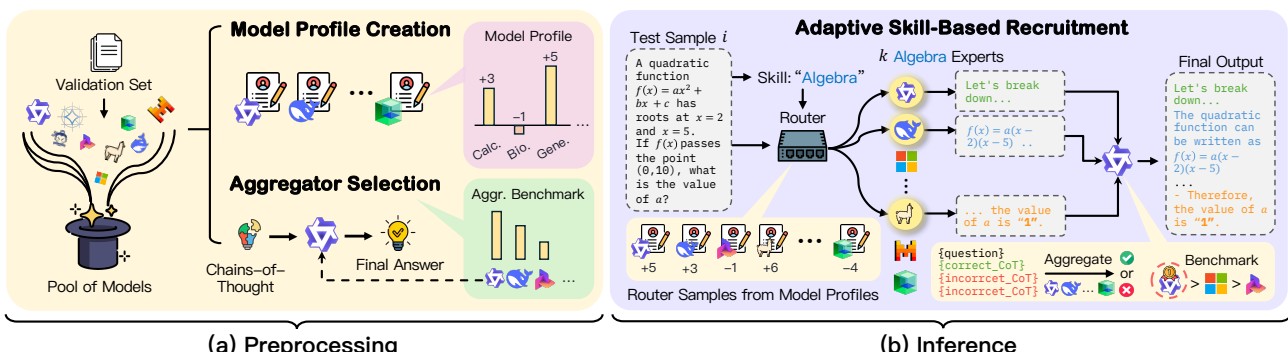

**(a) Preprocessing**       **(b) Inference**

*Figure 2.* Overview of SKILL-MOE. (a) Preprocessing: Given a validation set and a pool of agents, we create model profiles and select an aggregator. (b) Inference-Time: For each test example, SKILL-MOE activates the most relevant models (experts) based on skill-based routing, using model profiles determined during preprocessing. These models generate CoT responses, which the aggregator (chosen based on its ability to select correct answers) synthesizes into a final answer.

thesizing a final output with a predicted answer. We then benchmark each model's aggregation ability and select the best-performing aggregator for each task.

### 3.3. Inference

#### 3.3.1. SKILL-BASED RECRUITING

At inference time (see Figure 2 (b)), we follow the same keyword annotation procedure as in Appendix 3.2.1 to generate relevant keywords for the test sample. To align a test sample's keywords with those in the model profiles, we employ Sentence-BERT (Reimers & Gurevych, 2020) to match keywords via the cosine similarity between their embeddings. This ensures that every keyword gets matched to the closest counterpart in the model profile. Next, expert recruitment is performed by selecting the top $k$ models whose profiles best match the required skills of the test sample. This is determined by two factors: **(1) local suitability score** and **(2) global competency**. For each model $M_i$, its *local suitability score* for a test sample $q$, $\mathcal{S}(M_i, q)$ is computed as the sum of its skill scores over the set of keywords needed for $q$ (denoted as $K_q$). It can be expressed as follows:

$$\mathcal{S}(M_i, q) = \sum_{k_j \in K_q} s_{k_j}^{(i)}$$

where $s_{k_j}^{(i)}$ represents the score of model $M_i$ for the $j$-th skill in the test sample $q$. This results in an model ranking distribution $\mathcal{D}_q$ for each test sample $q$: $\mathcal{D}_q = \{\mathcal{S}(M_1, q), \mathcal{S}(M_2, q), ..., \mathcal{S}(M_n, q)\}$.

Intuitively, suppose $M_1$ has scores of $+3$, $+5$, and $-2$ for algebra, calculus, and geometry, respectively, which are needed for a given sample; its total score for this sample would be $3+5-2 = 6$. Calculating this score for all models yields a distribution of model strengths for the given sample, e.g., $\{M_1\colon 6, M_2\colon 3, ..., M_n\colon -10\}$, which is the ranking of *how suitable a model is for a sample*.

We also take into account each model's overall strength in a task, i.e., *global competency*. This is easily obtained by summing a model's score across all keywords in its profile, and normalizing it by the total sum of all models' scores. We denote this global strength as $\gamma_i$, representing a model's overall task performance *relative to others*. Finally, the expert selection is performed by sampling from the product of the local suitability score, $\mathcal{S}(M_i, q)$ (from a model rank distribution $\mathcal{D}_q$) and the global competency $\gamma_i$. That is, the relevance score of a model $M_i$ for a test sample $q$ is: $w_q^{(i)} = \gamma_i \mathcal{S}(M_i, q)$. We apply a softmax function with the temperature set to $0.5$ to this distribution $\{w_q^{(i)}\}_{i=1}^n$, and then sample $k$ experts with replacement for each problem, i.e.,

$$E_q^{(i)} \sim \text{Categorical}(w_q^{(1)}, w_q^{(2)}, ..., w_q^{(n)}), \ i = \{1, 2, ..., k\}$$

To enhance efficiency, we trim low-frequency experts, i.e., those who appear in fewer than $5\%$ of test cases. For example, given a test set with 100 samples, where 3 experts are recruited per sample (totaling 300 selections), any expert appearing fewer than $300 \times 5\% = 6$ times is discarded and resampled from the remaining higher-frequency experts. To visualize this, we provide the expert distribution before and after trimming in Figure 4.

#### 3.3.2. FINAL ANSWER GENERATION

After expert recruitment, each sample will be passed to the experts, i.e., the input for each expert is the test problem, $x_0 = q$. These experts generate their reasoning paths to the problem in the form of Chain-of-Thought (Wei et al., 2022): $y_0^{(i)} = E^{(i)}(x_0) \ \forall \ i \in \{1, 2, ..., k\}$. Then, the task-specific aggregator $A^*$ is introduced to synthesize the $k$ outputs into a high-quality final answer (Wang et al., 2024a). That is, the final answer is produced by: $y = A^*(\|_{i=1}^k y_0^{(i)})$, where $\|$ denotes the concatenation operation.

### 3.3.3. SCALABLE BATCHED INFERENCE

In our experiments, we mostly consider 7B–8B parameter LLMs, which have a substantial memory footprint. Due to the adaptive nature of the recruitment process, the set of participating LLMs may change dynamically for different problems. For instance, one sample may require Qwen, Llama, and Mistral, while another may need Gemma, Exaone, and Phi. A naive implementation of this approach can lead to high latency, particularly when the required models change frequently. To mitigate these computational challenges, we introduce a novel batching strategy to maximize throughput. Specifically, for a given set of instances, we precompute (using inferred skills) which LLMs will be called for each instance. We then group instances based on their required experts, as illustrated in Figure 3 (III) and Algorithm 1 in Algorithm 1. In other words, each active expert receives all assigned instances at once, ensuring that each expert is loaded only once per batch. This enables efficient batched inference on *a single GPU* while supporting a pool of 16 LLMs. Moreover, this approach is flexible, as more GPUs can further accelerate inference through parallelization.

## 4. Experimental Setup

### 4.1. Implementation Details

We use a model pool of 16 LLMs with sizes ranging from 3.5B to 12B parameters, with the majority falling in the 7–8B range. These include general-purpose instruction-tuned models, domain-specific fine-tuned variants on math and biology, and models distilled from DeepSeek R1's trajectories (DeepSeek-AI et al., 2025). A complete list of models is provided in Table 8. We measure performance on a diverse range of datasets, chosen to require expertise in a number of domains. First, we consider MMLU-Pro (Wang et al., 2024c), which is a harder version of MMLU (Hendrycks et al., 2021), containing a variety of questions across 14 college-level subjects. Given its large test set of 12,000 samples and the computational cost of evaluating proprietary models, we employ stratified sampling to create a subset of 2,100 samples, ensuring each category contains 150 samples. We also evaluate on AIME 2024, which is a challenging mathematics competition dataset containing math Olympiad problems. For more science-specific reasoning, we test on GPQA Diamond (Rein et al., 2023), which contains questions across a variety of science fields written by experts, explicitly designed to be difficult to answer even by skilled humans with web access. Finally, we include MedMCQA (Pal et al., 2022), which covers questions sourced from medical entrance exams across 21 medical subjects. For each dataset, we sample around 350 samples as

the validation set to create the model profiles.[2] Full dataset statistics are provided in Table 15, and the model pool we consider is shown in Appendix D.

### 4.2. Baselines

We compare against four categories of baselines.

- **Zero-shot single-model methods**: This category includes high-capacity open-source models like Qwen2.5 72B (Qwen et al., 2025) and Llama 3.3 70B (AI@Meta, 2024); and strong distilled 32B and 7B models such as QwenR1 (DeepSeek-AI et al., 2025). For reference, we also report the best task-specific model from our pool for each task, denoted as Task-Best.

- **Advanced single-model baselines with inference-time compute**: We evaluate methods that enhance inference-time reasoning, specifically Self-Refine (SR) (Madaan et al., 2023) and Self-Consistency (SC) (Wang et al., 2023b). To ensure a fair comparison, we set SC's sample size to 5, aligning with the number of large language model (LLM) calls in SKILL-MOE, which engages three experts and one aggregator model.[3] Additionally, for these baselines, we use the best-performing LLM for each task, inferred on the same validation set used for our agent profile creation.

- **Single-model multi-agent baselines**: To isolate the impact of SKILL-MOE's recruitment strategy, we compare against methods where multiple instances of the same model collaborate. Specifically, we consider Multi-Agent Debate (Debate) (Du et al., 2023) and Self-Mixture-of-Agents (Self-MoA) (Li et al., 2025), both of which rely on iterative, multi-round discussions using a single model. These baselines employ three agents, each using the same task-best model, and conduct two rounds of discussion, resulting in a total of 6 LLM calls per sample.

- **Multi-model multi-agent baselines**: We also evaluate approaches leveraging diverse models in a multi-agent setup. This includes Mixture-of-Agents (MoA) (Wang et al., 2024a) and ReConcile (Chen et al., 2024b), both of which incorporate a fixed set of models in multi-round interactions. To ensure a fair comparison with our approach, particularly in the use of the validation set, we select the top three performing models from the validation set and conduct multi-round interactions. In MoA, agents participate in two rounds of discussion, while agents in ReConcile engage in three rounds, leading to 6 and 9 LLM calls per sample, respectively.

---

[2]For AIME 2024, validation questions from prior years' problems (from 2012 to 2023).

[3]We use an odd number of SC calls to avoid ties.

*Table 1.* Comparison of SKILL-MOE with single-model and multi-model baselines. SKILL-MOE outperforms all multi-agent baselines and achieves performance comparable to strong proprietary models like GPT4o-mini, as well as 70B models, while primarily operating with 7-8B models. Notably, no single baseline consistently secures the second-best performance, even when the strongest models for each task are known. We **bold** the best results and underline the second-best (excluding bigger models, shown in gray).

| Category | Method | Model | AIME | MMLU-Pro | MedMCQA | GPQA | Avg. |
|---|---|---|---|---|---|---|---|
| Open-Source 30B+ Model | Zero-Shot CoT | Qwen2.5 72B | $25.55 \pm 3.85$ | $71.54 \pm 0.88$ | $69.02 \pm 0.32$ | $51.02 \pm 0.27$ | 54.28 |
| | Zero-Shot CoT | Llama3.3 70B | $32.22 \pm 3.85$ | $69.26 \pm 0.47$ | $59.78 \pm 0.74$ | $51.44 \pm 0.62$ | 53.18 |
| | Zero-Shot CoT | QwenR1 32B | $76.67 \pm 3.36$ | $64.43 \pm 0.31$ | $24.70 \pm 0.40$ | $61.95 \pm 0.77$ | 56.94 |
| Open-Source 7B Model | Zero-Shot CoT | QwenR1 7B | $55.93 \pm 5.16$ | $52.57 \pm 0.45$ | $38.72 \pm 0.44$ | $44.95 \pm 1.49$ | 48.04 |
| | Zero-Shot CoT | Task-Best | $55.93 \pm 5.16$ | $54.89 \pm 0.53$ | $55.44 \pm 0.50$ | $48.43 \pm 3.10$ | 53.62 |
| Advanced Single Model | Self-Refine (SR) | Task-Best | $53.33 \pm 3.34$ | $53.74 \pm 0.20$ | $49.57 \pm 0.59$ | $50.84 \pm 3.65$ | 51.87 |
| | Self-Consistency (SC) | Task-Best x5 | $\underline{67.78} \pm 1.57$ | $56.71 \pm 0.14$ | $56.85 \pm 0.11$ | $\underline{53.54} \pm 0.36$ | $\underline{58.72}$ |
| Single-Model Multi-Agent | Multi-Agent Debate | Task-Best x3 | $56.67 \pm 6.67$ | $56.21 \pm 0.55$ | $51.63 \pm 0.80$ | $50.51 \pm 0.51$ | 53.76 |
| | Self-MoA | Task-Best x3 | $55.56 \pm 5.09$ | $55.43 \pm 0.72$ | $53.27 \pm 0.60$ | $52.86 \pm 1.46$ | 54.28 |
| Multi-Model Multi-Agent | MoA | Top-3 | $41.11 \pm 5.09$ | $\underline{61.78} \pm 0.25$ | $59.29 \pm 0.32$ | $52.86 \pm 3.37$ | 53.76 |
| | ReConcile | Top-3 | $50.00 \pm 7.20$ | $56.46 \pm 0.10$ | $\mathbf{60.74} \pm \mathbf{0.43}$ | $47.98 \pm 2.32$ | 53.80 |
| | SKILL-MOE | Adaptive | $\mathbf{68.88} \pm \mathbf{5.08}$ | $\mathbf{63.71} \pm \mathbf{0.43}$ | $\underline{59.35} \pm 0.14$ | $\mathbf{57.78} \pm \mathbf{2.09}$ | $\mathbf{62.43}$ |

*Every baseline requiring **task-best model selections** is also based on its validation performance*, which is summarized in Table 9. Further details are provided in Appendix C.

# 5. Results and Analysis

## 5.1. Main Results

**SKILL-MOE consistently outperforms all baselines.** We present the main results in Table 1. Across all domains, SKILL-MOE shows superior performance compared to all baselines, beating single-model baselines (e.g., SR, SC) using the best overall model, multi-agent debate with a single model (e.g., Debate, Self-MoA), as well as multi-model multi-agent baselines (e.g., MoA, ReConcile). SKILL-MOE outperforms the most competitive multi-agent baseline, Self-MoA, by $8.15\%$ (absolute) on average, with consistent improvements across domains (e.g., $8.28\%$ gain on MMLU-Pro, $13.45\%$ on AIME, $4.92\%$ on GPQA, $6.08\%$ on MedMCQA). These gains are also seen when comparing to multi-model baselines like MoA and ReConcile that use the *top three strongest models* per domain. SKILL-MOE also substantially outperforms test-time scaling methods, such as SR (Madaan et al., 2023) and SC (Wang et al., 2023a). Surprisingly, with the task-best model, SC beats multi-agent debate baselines (e.g., Self-MoA, MoA), though it still underperforms SKILL-MOE by an average of $3.71\%$. This indicates that *if the task-best model is known,* scaling test-time compute is a simple yet effective way to improve performance, but adaptively selecting suitable experts at an instance level leads to even further improvements.

**SKILL-MOE generalizes well across tasks.** No single baseline in Table 1 is universally effective across all tasks. For instance, while MoA performs well on MMLU-

Pro, it struggles on AIME; ReConcile excels in MedM-CQA but fails to generalize to GPQA. Therefore, knowing which method works best for a task is nontrivial. In contrast, SKILL-MOE consistently delivers strong performance across all domains. Moreover, while SC with the task-best model is the most competitive setting on AIME and GPQA, it falls short on MMLU-Pro and MedMCQA, where multi-agent baselines perform better. This discrepancy may stem from the broader subject diversity in MMLU-Pro and MedMCQA, whereas AIME is more math-focused, and the task-best model, QwenR1 (DeepSeek-AI et al., 2025), delivers strong solo performance already. While QwenR1 demonstrates exceptional math and code reasoning capabilities ($55.93\%$ on AIME), leading to strong Self-Consistency performance ($67.78\%$), it struggles to generalize to other domains such as MedMCQA, highlighting the need for a robust and flexible framework like SKILL-MOE to generalize across diverse tasks.

**SKILL-MOE matches larger 32-72B models.** In Table 1, we also find that SKILL-MOE achieves performance that is on par with, and in several cases exceeds, strong open-source models in the 32-72B parameter range, despite operating with much smaller 7-8B models. To ensure a fair comparison, we include a strong thinking model, QwenR1 32B (DeepSeek-AI et al., 2025), which performs competitively on math- and science-heavy benchmarks such as AIME, MMLU-Pro, and GPQA, but exhibits markedly weaker performance on MedMCQA ($24.70\%$). This variability underscores a key limitation of single-model approaches: *strong performance typically depends on prior knowledge of which model is best suited for each task.* In contrast, SKILL-MOE requires no prior assumptions about the task, effectively leveraging each model's strengths at an instance level, and achieving consistent performance across all benchmarks,

*Table 2.* Accuracy on **OmniMATH**, with the model profiles from MMLU-Pro and AIME. Improvements (+x%) are shown in absolute gain, compared to the best baseline.

| | Model | Profile From | |
|---|---|---|---|
| | | **MMLU-Pro** | **AIME** |
| **Omni Acc.** | Debate | 34.51 | 42.93 |
| | MoA | 31.55 | 47.36 |
| | Self-MoA | 14.63 | 48.75 |
| | ReConcile | 22.01 | 42.62 |
| | SKILL-MOE | **49.32 (+14.81)** | **52.03 (+3.28)** |

*Table 3.* Ablations on different aggregators in our full setting. While adaptively choosing an aggregator at the instance level is feasible, it empirically underperforms task-level selection.

| Aggregator | MMLU-Pro | GPQA |
|---|---|---|
| Random | 52.29 | 48.92 |
| Adaptive | 57.12 | **58.01** |
| Task-specific | **63.71** | 57.78 |

resulting in the highest overall average accuracy (62.43%) even compared to larger models.

### 5.2. Additional Analysis

**SKILL-MOE generalizes to unseen tasks.** Given the constant introduction of new, unseen data in LLM deployments, we evaluate whether SKILL-MOE can generalize using existing model profiles on an additional benchmark: Omni-MATH (Gao et al., 2025), an Olympiad-level math dataset with 4.4k test samples. In Table 2, we reuse model profiles constructed from MMLU-Pro and AIME's validation set, and directly test on OmniMATH. All multi-agent baselines also use top-3 models selected from the same validation sets. When using the MMLU-Pro profile, multi-agent baselines struggle on OmniMATH because MMLU-Pro is broad and diverse, whereas OmniMATH is highly math-focused. This domain discrepancy, combined with the fact that the baselines rely on a fixed set of task-best models, leads to a 14.81% accuracy gap between the best baseline (Debate) and SKILL-MOE. In contrast, SKILL-MOE mitigates the profile-domain gap by leveraging fine-grained skill representations: rather than committing to a fixed model set, it dynamically recruits experts at the instance level based on the specific skills required. As skills transfer more readily across domains than task-specific model selections, SKILL-MOE is more robust to domain shift. Switching to the AIME profile allows the baselines to leverage math-specialized models, substantially improving their performance. Nonetheless, SKILL-MOE's instance-level recruitment still surpasses the strongest baseline (Self-MoA) by 3.28%.

**Role and selection of the aggregator.** Unlike most of our discussion-based multi-agent baselines, SKILL-MOE

*Table 4.* Comparison of different expert selection and aggregation strategies. The best two combined lead to the best result.

| Expert | Aggregator | GPQA |
|---|---|---|
| Random | Task-Specific | 31.82 |
| Recruited | Random | 51.52 |
| Recruited | Majority Vote | 53.54 |
| Recruited | Task-Specific | **57.78** |

collects a single CoT and answer from each expert and combines them via an aggregator. This provides efficiency gains, as shown in Table 6; here, we investigate the role of the aggregator in our framework. While experts are selected per instance, the aggregator is chosen per task, as we find that reasoning ability does not necessarily translate to effective aggregation. Table 3 compares three strategies: (1) a randomly chosen aggregator from the model pool (**Random**), (2) an instance-level aggregator selection based on model profiling (**Adaptive**), and (3) a task-specific aggregator determined by task-level performance (**Task-specific**). Evaluated on MMLU-Pro and GPQA, the results indicate that a random aggregator substantially degrades performance, showing that the aggregator plays a crucial role. While the instance-level aggregator improves outcomes on both datasets, the task-specific aggregator outperforms it on MMLU-Pro and performs comparably on GPQA. We further find that the similar performance of instance-specific and task-specific aggregation on GPQA is due to a high degree of overlap in selected aggregators. Overall, this suggests that good reasoners will not always be good aggregators, supporting our task-based selection.

**Synergy between expert and aggregator selection.** As demonstated that the selection of an aggregator plays an important role, we further investigate the synergy between good aggregator and good expert selection. In Table 4, we experiment with two expert selection strategies: (1) randomly recruiting $k$ experts without using model profiles, and (2) using our model profiles to recruit experts. We also employ three settings for aggregator selection: (1) based on the task performance as SKILL-MOE uses, (2) randomly selecting an aggregator, and (3) using majority vote to select the final answer without using any aggregator. Our findings indicate that the combination of strong models with a task-specific aggregator yields the highest performance. When the aggregator is suboptimal, majority voting can serve as a robust alternative. However, when the expert models themselves are weak (chosen randomly), even a strong aggregator cannot fully compensate for the performance drop. To further understand the source of improvements and disentangle expert quality from aggregation quality, we conduct additional experiments in Appendix M and find that (1) given the same set of experts, the chosen aggregator outperforms majority vote, (2) when using a majority vote, our recruited

*Table 5.* Comparison of different recruiting strategies on GPQA. Compared to using the identified task-fixed top models, using our model profile at the instance level performs better.

| Recruiting Strategy | Acc. |
|---|---|
| Top-3 Experts | 52.86 |
| Top-5 Experts | 47.68 |
| 3 Random Experts | 42.61 |
| 5 Random Experts | 44.92 |
| Model Profile (Ours) | **57.78** |

$k$ experts surpasses repeatedly calling the task-best model $k$ times. These suggest that both expert selection and aggregator selection are important and mutually reinforcing.

**Utility of model profile.** SKILL-MOE profiles models based on their skills and leverages this information to recruit experts effectively. To underscore the importance of this step, we compare several alternative selection strategies in Table 5, evaluating accuracy on GPQA. In the top-$k$ approach, experts are fixed as the best-performing models for the task, whereas in the random-$k$ strategy, the selected experts vary across instances. Our results demonstrate that skill-based selection is essential. Notably, although selecting the top $k$ experts for a task may seem intuitive, it consistently underperforms compared to SKILL-MOE's adaptive instance-level expert selection. Interestingly, top-5 selection performs worse than top-3 selection, suggesting that a broader selection increases the likelihood of including weaker models, and that contributes to performance degradation. Additionally, the random selection strategy consistently harms performance, showing a $12.86\%$ to $15.61\%$ drop compared to SKILL-MOE, likely also due to the inclusion of weaker experts.

### 5.3. Efficiency Anaylsis

**Run time efficiency.** Enabling our batch inference strategy (Figure 3) substantially reduces inference time. We evaluate average run-time on GPQA and compare SKILL-MOE to a naive sequential implementation and to MoA (Table 6). As expected, sequential inference yields the highest latency, since the model must be repeatedly loaded and offloaded for each instance. In contrast, SKILL-MOE achieves $44\%$ lower latency than MoA on a single GPU—while also delivering higher accuracy. Notably, SKILL-MOE on a single GPU matches the run-time of MoA on 4 GPUs, and when scaled to 4 GPUs, SKILL-MOE attains nearly a $2\times$ speedup over MoA. We also report token usage in Appendix I, showing that SKILL-MOE achieves the best trade-off between efficiency and performance among multi-agent baselines. While our batch inference strategy does assume that test samples arrive in batches (so expert assignments can be predetermined), this assumption is not restrictive in practice: many widely-used inference acceleration techniques make

*Table 6.* Efficiency comparison of MoA and SKILL-MOE on the GPQA test set. Run time is averaged per sample.

| Method | # GPUs | Run Time (s) |
|---|---|---|
| Sequential | 1 | 196.92 |
| MoA | 1 | 45.98 |
| MoA | 4 | 21.66 |
| SKILL-MOE | 1 | 25.76 |
| SKILL-MOE | 4 | **10.85** |

*Table 7.* While multi-round discussion stabilizes performance with suboptimal aggregators, it *has little effect* with an optimal aggregator.

| Discuss | Aggr. | MMLU-Pro | GPQA |
|---|---|---|---|
| ✓ | Adaptive | 59.07 | 57.01 |
| ✗ | Adaptive | 57.12 | **58.01** |
| ✓ | Task-best | 57.81 | 57.78 |
| ✗ | Task-best | 56.67 | 57.01 |
| ✓ | Task-specific | **63.83** | 57.72 |
| ✗ | Task-specific | 63.71 | 57.78 |

the same assumption (e.g, vLLM (Kwon et al., 2023)), and popular services like ChatGPT (OpenAI, 2025) and Gemini (Gemini, 2025) also support batched inference.

**Methodological efficiency.** Like multi-agent discussion baselines, SKILL-MOE can also operate in a discussion-based manner. Instead of immediately aggregating initial responses, models first engage in three rounds of discussion—observing each other's outputs and generating refined responses—before submitting them to the aggregator. Table 7 reports results for this setting, comparing the adaptive aggregator (suboptimal), the task-best aggregator (suboptimal), and the task-specific aggregator (optimal) on MMLU-Pro and GPQA. With the task-specific aggregator, discussion yields only marginal gains on MMLU-Pro (63.83 vs. 63.71) and a slight drop on GPQA (57.72 vs. 57.78). While round-wise discussion can improve performance incrementally, the final outcome is primarily determined by aggregator strength. Consequently, SKILL-MOE with a strong task-specific aggregator can skip the costly multi-round discussion, reducing runtime and improving methodological efficiency, while surpassing the performance of discussion-based baselines, as shown in Table 6.

### 5.4. Discussion

A key feature highlighted in Table 1 is the *consistency* of SKILL-MOE's performance. While baseline methods occasionally do well in isolated settings (e.g., MoA on MMLU-Pro, ReConcile on MedMCQA), it is important to highlight that no baseline does well consistently across settings. This means that – without SKILL-MOE – getting a strong overall result would require evaluating all the baseline methods

and choosing the best settings manually. In contrast, SKILL-MOE achieves high performance by automatically recruiting the experts based on skills needed for each instance, serving as a robust recipe that generalizes across domains.

**Modularity.** Another key advantage of SKILL-MOE is its modularity. Unlike typical Mixture-of-Experts (MoE) frameworks, which need to be trained end-to-end from scratch in a centralized manner, SKILL-MOE uses the symbolic output channel of existing models to combine experts. This gradient-free approach enables seamless integration of pre-trained models without updates, allowing them to be trained independently and distributedly. Moreover, while standard MoEs have a fixed size determined before training, SKILL-MOE can *adaptively grow and evolve* as models are updated. Given the rapid advancements in LLMs, state-of-the-art models are often replaced within months. SKILL-MOE's modular and gradient-free design simplifies incorporation of these updates, requiring only a few calls to obtain a new model's profile, which can then be easily plugged into this framework. It is also straightforward to increase the number of experts recruited at test time, which is usually fixed in the typical MoE setting.

## 6. Conclusion

We introduced SKILL-MOE, a scalable MoE framework that coordinates models through symbolic outputs (i.e., natural language discussion). SKILL-MOE identifies the skills required for a given problem, recruits the corresponding experts, and aggregates their responses into a single high-quality answer. Across four diverse reasoning datasets, SKILL-MOE consistently outperforms inference-time scaling methods, debate frameworks, and recent mixture-of-agents approaches, delivering strong and stable performance even as the best baseline varies across domains. Its average performance on heterogeneous tasks also matches advanced proprietary systems such as GPT-4o-mini and larger 70B-parameter models. Detailed analysis shows that both expert selection and the choice of aggregator are crucial for downstream performance, and with a strong aggregator, costly multi-round discussions can often be skipped without sacrificing quality. Finally, SKILL-MOE introduces a novel batching strategy that runs efficiently on a *single GPU*, while retaining the flexibility to scale further with additional GPUs—achieving both high performance and efficiency.

## Impact Statement

In this work, we propose an inference-time method, SKILL-MOE, which operates without the need for additional training or fine-tuning. Consequently, the LLMs utilized by SKILL-MOE may still exhibit stereotypes, biases, and other negative traits inherent in their pre-training data (Weidinger

et al., 2021), over which we have no control. Therefore, the outputs produced by SKILL-MOE carry the same potential for misuse as those from other test-time methods. Further research is necessary to assess and mitigate these biases.

## Acknowledgements

This work was supported by NSF-CAREER Award 1846185, DARPA ECOLE Program No. HR00112390060, Microsoft Accelerate Foundation Models Research (AFMR) grant program, NSF-AI Engage Institute DRL-2112635, National Institutes of Health (NIH) under other transactions 1OT2OD038045-01, and Cisco and Capital One Faculty Awards. The views and conclusions contained in this document are those of the authors and should not be interpreted as representing official policies, either expressed or implied, of the NIH or other sponsors.

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

# Appendix

## A. The Use of Large Language Models (LLMs)

We use ChatGPT[4] for grammar correction and refinement. The model was only used to polish text written by the authors and was not used to contribute to research ideation or the generation of original content.

## B. Illustrations of Variations in Batch Inference

As discussed in Appendix 3.3, there are mainly three different ways to serve multiple LLMs to solve every instance adaptively. We illustrate these variations in Figure 3. Our batch inference method allows for the speedups featured in Table 6.

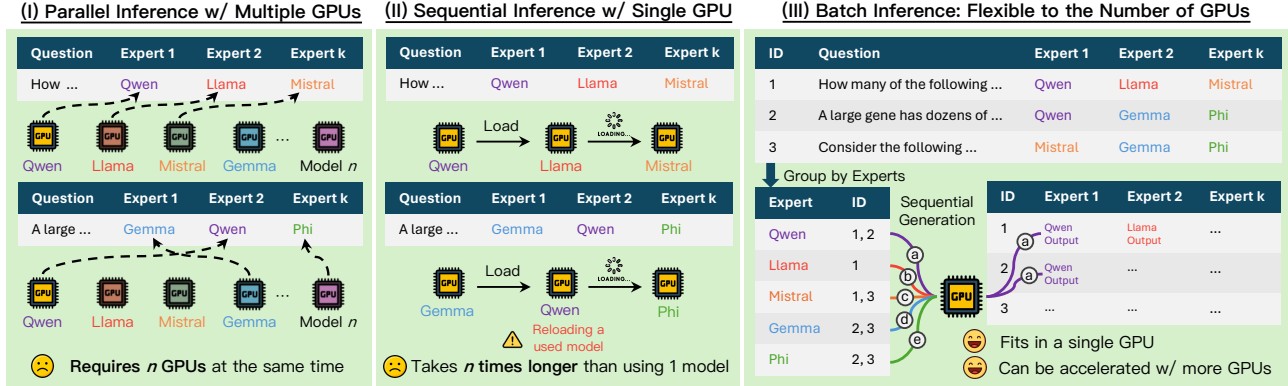

*Figure 3.* Different approaches to achieving adaptiveness in SKILL-MOE, which uses different models for each instance. In a naive setup (I), $k$ GPUs must be hosted simultaneously, allowing immediate access to outputs from each model. Another naive setup (II) requires only a single GPU but involves constant loading and offloading of models to obtain outputs from the corresponding model. Our scalable batch inference process (III) strikes a balance between (I) and (II). When models are assigned to problems, we group samples by model and sequentially load the corresponding LLM onto a single GPU to generate outputs efficiently. Moreover, this approach still allows us to parallelize across GPUs if they are available.

## C. Implementation Details

We conduct our experiments for SKILL-MOE and other single-model baselines on a single A6000 GPU with 48 GB of memory, while MoA and ReConcile are executed on 8 A6000 GPUs for parallelization. For the 70B models, we use the original version without quantization and perform inference on 4 A6000 GPUs. All open-source models utilize vLLM (Kwon et al., 2023) for inference. The temperature is set to 0.7 for all methods. The maximum output token length is fixed at 4096 for all models, except for QwenR1 and LlamaR1, which have a limit of 32768 since they are trained with longer trajectories and tend to generate longer outputs. All results, except those from proprietary models (due to budget constraints), are averaged over three random seeds. Further details on the model pool, distribution of the expert recruited, and all the prompts we use can be found in Table 8 and Appendix R.

## D. Model Pool

We provide the full list of our model pool in Table 8, including their names, sizes, and publicly available checkpoints on Huggingface. Most of the model sizes are 7 to 8 billion.

## E. Performance on the Validation Set

Table 9 shows the performance of each model on the validation set. We highlight the top-1 and top-3 models in bold font and yellow background, respectively. This information is also used for the baselines we compare against in Table 1.

---

[4]https://chatgpt.com/

*Table 8.* The models constituting the model pool.

| Model Name | Size | Huggingface Link |
|---|---|---|
| BioLlama | 8B | ContactDoctor/Bio-Medical-Llama-3-8B |
| DeepSeekMath | 7B | deepseek-ai/deepseek-math-7b-instruct |
| Exaone | 7.8B | LGAI-EXAONE/EXAONE-3.5-7.8B-Instruct |
| Gemma2 | 9B | google/gemma-2-9b-it |
| GLM4 | 9B | zai-org/glm-4-9b-chat |
| Granite | 8B | ibm-granite/granite-3.1-8b-instruct |
| InternLM3 | 8B | internlm/internlm3-8b-instruct |
| Llama3.1 | 8B | meta-llama/Llama-3.1-8B-Instruct |
| LlamaR1 | 8B | deepseek-ai/DeepSeek-R1-Distill-Llama-8B |
| Mathstral | 7B | mistralai/Mathstral-7B-v0.1 |
| Mistral | 12B | mistralai/Mistral-Nemo-Instruct-2407 |
| Phi3.5-mini | 3.5B | microsoft/Phi-3.5-mini-instruct |
| Qwen2.5 | 7B | Qwen/Qwen2.5-7B-Instruct |
| Qwen2.5-Coder | 7B | Qwen/Qwen2.5-Coder-7B-Instruct |
| Qwen2.5-Math | 7B | Qwen/Qwen2.5-Math-7B-Instruct |
| QwenR1 | 7B | deepseek-ai/DeepSeek-R1-Distill-Qwen-7B |

*Table 9.* Comparison of model performance on the validation set. The best model on each task is **bolded**, and the top 3 models on each task are highlighted in yellow.

| Model | MMLU-Pro | AIME | GPQA | MedMCQA |
|---|---|---|---|---|
| BioLlama | 37.71 | 0.85 | 27.31 | 42.86 |
| DeepSeekMath | 32.57 | 3.32 | 28.11 | 35.71 |
| Exaone | 52.29 | 25.99 | 32.13 | 56.35 |
| Gemma | 53.71 | 7.73 | 36.95 | 64.29 |
| GLM | 50.29 | 7.37 | 30.92 | 58.33 |
| Granite | 43.43 | 5.92 | 34.14 | 56.15 |
| InternLM | 43.14 | 7.91 | 36.14 | 55.56 |
| Llama | 46.00 | 6.78 | 33.73 | 66.87 |
| LlamaR1 | **54.29** | 51.98 | **56.22** | 53.37 |
| Mathstral | 34.57 | 3.11 | 36.55 | 52.38 |
| Mistral | 45.14 | 1.41 | 33.73 | 46.43 |
| Phi | 46.57 | 1.41 | 47.79 | 65.87 |
| Qwen | 54.00 | 13.56 | 37.35 | **67.06** |
| QwenCode | 46.29 | 9.89 | 30.52 | 50.79 |
| QwenMath | 31.71 | 11.13 | 28.51 | 36.90 |
| QwenR1 | 53.43 | **57.06** | 51.41 | 37.90 |

## F. Performance of Each Model as an Aggregator

Table 10 shows the performance of each model when acting as an aggregator. Note that the best-performing model in Table 9 can be different from the best aggregator model in Table 10, motivating us to choose the aggregator based on this synthetic task described in Appendix 3.2.2.

*Table 10.* Performance of each model when used as an aggregator, on the validation set. The best model on each task is **bolded**, and is selected as the task-specific aggregator.

| Model | MMLU-Pro | AIME | GPQA | MedMCQA |
|---|---|---|---|---|
| BioLlama | 37.31 | 21.47 | 30.12 | 42.46 |
| DeepSeekMath | 32.57 | 5.37 | 21.69 | 35.71 |
| Exaone | 57.43 | 47.92 | 35.34 | 52.58 |
| Gemma | 49.71 | 3.11 | 31.73 | 53.37 |
| GLM | 52.57 | 26.27 | 35.34 | 51.39 |
| Granite | 48.86 | 36.44 | 38.96 | 48.02 |
| InternLM | 55.14 | 16.95 | 42.57 | 51.59 |
| Llama | 51.14 | 11.86 | 40.56 | 50.60 |
| LlamaR1 | **59.71** | 53.67 | 46.18 | 49.01 |
| Mathstral | 41.71 | 26.27 | 35.74 | 46.43 |
| Mistral | 48.00 | 18.93 | 33.33 | 46.43 |
| Phi | 27.71 | 9.04 | 26.10 | 25.40 |
| Qwen | 56.86 | 38.14 | 39.36 | **53.37** |
| QwenCode | 51.14 | 29.66 | 38.96 | 50.79 |
| QwenMath | 31.71 | 5.93 | 16.06 | 36.90 |
| QwenR1 | 58.00 | **57.63** | **48.59** | 45.44 |

## G. Distribution of Experts

We present the distribution of recruited experts across different datasets in Figure 4. As noted in Appendix 3.2.1, we trim experts with occurrences below 5% to reduce model loading time. In Figure 4, the top row shows the distribution before trimming, and the bottom row shows the distribution after trimming. The distribution varies significantly across datasets – on more diverse datasets such as MMLU-Pro, the recruited experts are also more varied. In contrast, for AIME and GPQA, which focus more on math and science, the recruited experts are dominated by a few models.

## H. Distribution of Skills

In practice, the extracted skills are interpretable and align well with human intuition. We provide top-5 skills for AIME 2024 and MMLU-Pro in Table 11. AIME is dominated by core math areas such as algebra, number theory, and geometry. MMLU-Pro exhibits a long-tail distribution over diverse domains (e.g., psychology, finance, law), reflecting its broad coverage. This indicates that the labels capture meaningful high-level structure rather than noise.

## I. Test-time Token Count Analysis

In addition to measuring GPU run time in Table 6, we compare the test-time token count with multi-agent baselines. As shown in Figure 5, SKILL-MoE uses fewer tokens than Self-MoA while achieving a significant performance gain. However, compared to MoA and ReConcile, SKILL-MoE generates more tokens, particularly on GPQA. The primary reason, as illustrated in Figure 4, is that SKILL-MoE predominantly recruits LlamaR1 and

*Table 11.* Preview of the skill distributions.

| Skill | Count | Percent |
|---|---|---|
| **AIME 2024 (Unique keywords: 75)** | | |
| Algebra | 15 | 9.43% |
| Number Theory | 13 | 8.18% |
| Geometry | 12 | 7.55% |
| Combinatorics | 9 | 5.66% |
| Trigonometry | 6 | 3.77% |
| **MMLU-Pro (Unique keywords: 1053)** | | |
| Psychology | 185 | 1.66% |
| Finance | 184 | 1.65% |
| Economics | 159 | 1.42% |
| Thermodynamics | 134 | 1.20% |
| Law | 112 | 1.00% |

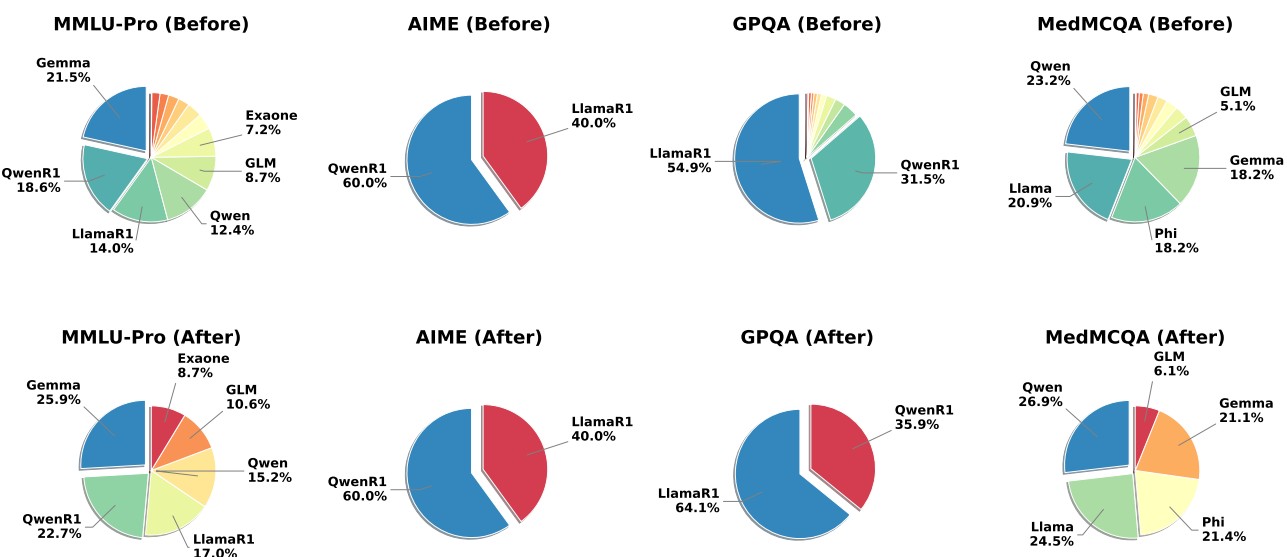

*Figure 4.* Distribution of the recruited experts across datasets. Top row: the distribution before trimming. Bottom row: the distribution after trimming and resampling.

QwenR1, both of which are trained with long reasoning trajectories, resulting in substantially longer outputs compared to other models. This explains why SKILL-MOE requires less GPU run time despite producing more tokens: by skipping the expensive multi-round discussions, we eliminate the time spent loading and offloading models. However, the inherent verbosity of the R1 models contributes to the higher token count.

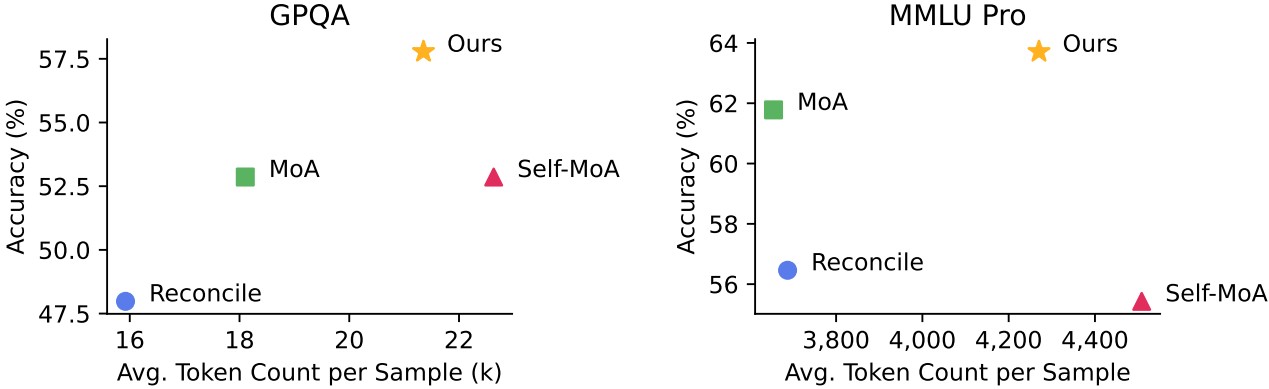

*Figure 5.* Comparison of the test-time token count used in different methods.

## J. The Effect of Trimming and Re-sampling

As described in Appendix 3.3.1, we trim the recruited experts if their occurrence falls below $5\%$ of the total selections. Here, we analyze the impact of this trimming process. Without trimming, the diversity of selected experts increases, but the model loading time also increases. Table 12 presents a quantitative comparison of accuracy and GPU run time using 4 GPUs. As expected, trimming reduces GPU run time across both datasets by minimizing the need to load infrequently used models. Interestingly, we also observe that trimming improves accuracy on GPQA. This improvement may be due to the fact that after trimming, only

*Table 12.* Trimming the low-frequent experts improves both accuracy and efficiency.

|  | MMLU-Pro | | GPQA | |
|---|---|---|---|---|
|  | Acc ↑ | Time ↓ | Acc ↑ | Time ↓ |
| No Trim | **63.94** | 18.83 | 55.26 | 21.78 |
| w/ Trim | 63.71 | **12.27** | **57.78** | **10.85** |

*Table 14.* Comparison of aggregation strategies on MMLU-Pro and GPQA.

| Setting | MMLU-Pro | GPQA |
|---|---|---|
| Task-Best ×3 + Majority Vote | 58.39 | 53.34 |
| Task-Best ×3 + Our Aggregator | 61.22 | 54.40 |
| Recruited Experts + Majority Vote | 60.71 | 53.34 |
| Recruited Experts + Our Aggregator (SKILL-MOE) | **63.71** | **57.78** |

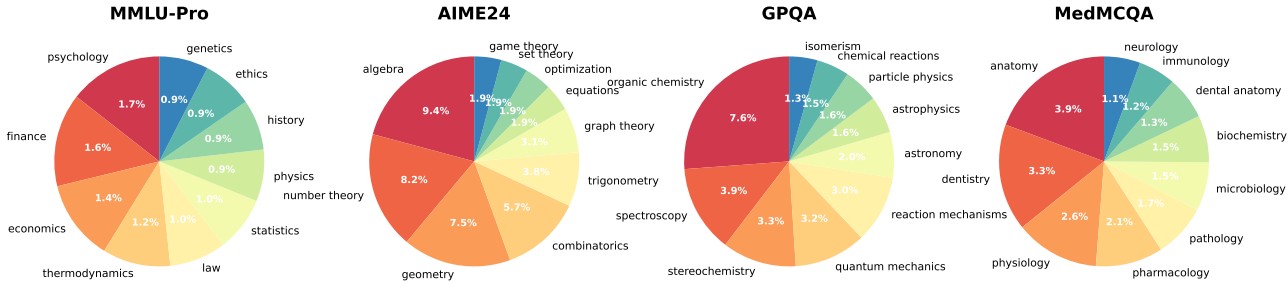

*Figure 6.* Keyword distribution in the validation set. For brevity, we show only the top 10 keywords.

LlamaR1 and QwenR1 remain as the recruited experts, which are particularly effective on this task.

## K. Generalization to Less Heterogeneous Tasks

We additionally evaluate on ARC-Challenge (Clark et al., 2018), a multi-step commonsense reasoning benchmark with less explicit domain structure. We use the 299 samples from the official validation set. Results in Table 13 show that even in a setting where the task is less heterogeneous, our method still achieves the best performance among the main baselines (although the margin is smaller than what we observe on the more heterogeneous benchmarks in Table 1).

*Table 13.* Performance comparison on ARC-Challenge.

| Method | Acc. (%) |
|---|---|
| Task-Best Model | 88.6 |
| Self-Consistency (Best ×5) | 89.3 |
| Mixture-of-Agents | 90.1 |
| ReConcile | 89.0 |
| SKILL-MOE | **91.7** |

## L. Analysis of Failure Modes, Domain Bias, and Robustness

SKILL-MOE generalizes well across diverse domains (math, STEM, medical, general, and commonsense) and robustly handles noisy inputs, outperforming standard majority voting. However, the model remains susceptible to two primary failure modes: collective expert failure, where no expert produces a correct answer (occurring in 33.3% of AIME and 29.3% of GPQA evaluations), and aggregation errors, where the aggregator selects an incorrect response (accounting for 20.0% of AIME and 18.7% of GPQA errors).

## M. Disentangling Expert Quality from Aggregation Quality

To further understand the source of improvements and disentangle expert quality from aggregation quality, we additionally evaluated applying the task-selected aggregator on top of three calls to the task-best model (i.e., replacing the majority vote in self-consistency). By comparing the first two rows, we found that using the task-best model three times, combined with the chosen aggregator, outperforms simple majority vote. Moreover, recruiting $k$ experts using our model profiles also surpasses repeatedly calling the task-best model $k$ times when aggregation is done via majority vote (Row 3 vs. Row 1). When both expert selection and aggregator selection are combined, we obtain the best overall performance, highlighting that both components are important and mutually reinforcing.

## N. Keyword Distribution in Validation Data

We provide the annotated keyword distribution in Figure 6.

## O. Dataset Statistics and Licenses

We provide the sample sizes and licenses of the datasets used in this work in Table 15. All the datasets are in English, and all datasets are used in a fashion consistent with their intended use.

*Table 15.* The statistics and licenses of the datasets we use in this work.

|  | Validation Size | Test Size | License |
|---|---|---|---|
| MMLU-Pro (Wang et al., 2024c) | 350 | 2,100 | Apache License |
| AIME (MAA, 2024) | 354 | 30 | CC0 |
| GPQA (Rein et al., 2023) | 249 | 198 | MIT License |
| MedMCQA (Pal et al., 2022) | 504 | 4,183 | MIT License |

## P. Sensitivity to the keyword LLM

We choose Qwen 2.5 7B (Qwen Team, 2024) as the "Keyword LLM" to generate the required skills for each instance during both preprocessing and inference. Here, we investigate the sensitivity of the results to the choice of the Keyword LLM, testing three different models: Qwen 2.5 7B (Qwen Team, 2024), Llama 3.1 8B (AI, 2024), and Gemma 2 9B (Team et al., 2024). As shown in Table 16, the final performance remains consistent regardless of the chosen model, indicating that the selection of the Keyword LLM has minimal influence on performance.

*Table 16.* Keyword LLM has little effect on the final performance.

| Keyword LLM | MMLU-Pro | GPQA |
|---|---|---|
| Llama 3.1 8B | 64.19 | 56.62 |
| Gemma 2 9B | 64.02 | 57.01 |
| Qwen 2.5 7B | 63.71 | 57.78 |

## Q. Algorithm

We provide the algorithm for our batched inference strategy in Algorithm 1.

---

**Algorithm 1** Batched Inference

---

**Require:** Test samples $\mathcal{Q}$, model pool $\mathcal{M}$
**Ensure:** Inference results for all samples
1: *expert_sample_map* $\leftarrow \emptyset$ {Map from expert to assigned samples}
2: **for all** $q \in \mathcal{Q}$ **do**
3:    $E_q \leftarrow$ RECRUITEXPERTS$(q, \mathcal{M})$ {Select $k$ experts per sample (Section 3.3.1)}
4:    **for all** $e \in E_q$ **do**
5:       *expert_sample_map*$[e] \leftarrow$ *expert_sample_map*$[e] \cup \{q\}$
6:    **end for**
7: **end for**
8: *results* $\leftarrow \emptyset$ {Aggregate inference outputs}
9: **for all** $(e, q_e) \in$ *expert_sample_map* **do**
10:    *results* $\leftarrow$ *results* $\cup$ $e$.GENERATE$(q_e)$ {Batched inference per expert}
11: **end for**
12: **return** *results*

---

## R. Prompts

---

**Prompt for the Keyword LLM to Generate Keywords**

Question: {question}

What are the core knowledge, subjects or skills needed to solve this problem? List 2-5 keywords separated in comma. Example keywords: psychology, virology, behavioral theory, microbiology, diplomacy, political science, property law, finance, business. Give ONLY the keywords, no other words or explanation.

Follow this format: Keywords: <keyword1>, <keyword2>...

---

**Prompt for Zero-shot Chain-of-Thought Generation (Multiple Choice)**

Question: {question}

Provide your step-by-step reasoning first, and then print "The answer is (X)" where X is the answer choice (one capital letter), at the end of your response.

---

**Prompt for Zero-shot Chain-of-Thought Generation (Math)**

Question: {question}

Provide your step-by-step reasoning first, and then print "The answer is \\boxed{X}", where X is the final answer, at the end of your response.

---

**Prompt for the Aggregator (Wang et al., 2024a)**

You have been provided with a set of responses from various open-source models to the latest user query. Your task is to synthesize these responses into a single, high-quality response. It is crucial to critically evaluate the information provided in these responses, recognizing that some of it may be biased or incorrect. Your response should not simply replicate the given answers but should offer a refined, accurate, and comprehensive reply to the instruction. Ensure your response is well-structured, coherent, and adheres to the highest standards of accuracy and reliability.

Responses from models:

{model_1_response}

{model_2_response}

{model_3_response}

Question: {question}

Provide your step-by-step reasoning first, and then print "The answer is (X)" where X is the answer choice (one capital letter), at the end of your response.

---

# S. SKILL-MOE as a Sparse Mixture-of-Expert

We use 'Mixture-of-Experts' in a functional sense: selectively recruiting and combining specialized experts per input. Unlike conventional MoE, where experts are parameter subsets within a single model, our experts are independently pre-trained LLMs. They are not jointly trained and do not share parameters. Instead, we route at the model level based on inferred skills and aggregate via natural language. Our approach is thus a symbolic, training-free MoE framework. In the Sparse Mixture-of-Experts (SMoE) framework (Shazeer et al., 2017a), a trainable router dynamically selects a subset of experts for each input. Formally, given an input $x$, the output of an SMoE layer, $y$ is computed as:

$$y = \sum_{i=1}^{k} \mathcal{R}(x)_i \cdot f_i(x),$$
$$\mathcal{R}(x) = \text{softmax}(\text{Top-K}(g(x)), k) \tag{1}$$

where $f_i(x)$ represents the response of the $i$-th expert, and $\mathcal{R}(x)$ is a trainable router that assigns selection probabilities to each expert based on $g(x)$, typically a small feedforward network (Shazeer et al., 2017b; Riquelme et al., 2021). The Top-K operation retains only the top $k$ experts, setting the probabilities of others to zero after the softmax operation.

However, directly applying SMoE in our framework presents key challenges. Unlike SMoE, our method operates in a symbolic, text-based space and is designed for test-time inference, meaning that we do not rely on a trainable router to learn expert selection, nor do the experts in our method refer to model parameters. Instead, we introduce a skill-based routing mechanism to select relevant experts based on predefined competencies rather than learned gating functions. Formally, our aggregation process can be expressed as:

$$y = A^*(\|_{i=1}^{k} y^{(i)})$$
$$y^{(i)} = E^{(i)}(x) \, \forall \, i \in \{1, 2, ..., k\} \tag{2}$$
$$E^{(i)} \sim \text{Categorical}(w^{(1)}, w^{(2)}, ..., w^{(n)}) \, \forall i \leq k$$

where $A^*$ is the aggregator model determined via validation set, and $\|$ denotes the concatenation of experts' responses, i.e., $y^{(\cdot)}$. Here, $y^{(j)}$ represents the output of expert $j$'s forward response given an input $x$, defined as $E^{(j)}(x)$. Each expert $E^{(i)}$, $\forall i \leq k$ is selected from our proposed skill-based routing strategy (Section 3.3.1). In short, we construct model profiles using a validation set to evaluate each model's specialization across different skills. This allows us to estimate a probability distribution $w^{(j)}$ over models based on both their suitability for the required skills and their global competence relative to other experts.

This skill-based routing framework retains the core benefits of SMoE while removing the reliance on a trainable gating mechanism. Specifically, the aggregator model $A^*$ in SKILL-MOE plays a role analogous to the weighted sum ($\sum$) operation in SMoE, synthesizing outputs from selected experts. Likewise, the recruited agent $E^{(i)}$ corresponds to the Top-k operation in SMoE, ensuring that only the most relevant and specialized experts contribute to the final output. We inherit the key conceptual benefits of SMoE – dynamic expert selection and response aggregation – while also introducing additional advantages. SKILL-MOE is gradient-free, eliminating the need for retraining, and is entirely automatic, leveraging a large pool of pre-trained models to deliver a better performance.

