# OpenReview forum: "Skill-Based Mixture-of-Experts: Adaptive Routing for Heterogeneous Reasoning via Inferred Skills"
_ICML.cc/2026/Conference — ICML 2026 regular_

### Official Review · Reviewer_tpjS · 2026-02-25

**Soundness:** 3
**Presentation:** 3
**Significance:** 2
**Originality:** 2
**Overall Recommendation:** 4
**Confidence:** 4

**Summary:**

This work proposes symbolic MoE, a way to select a subset of LLMs and aggregate their responses. Results with symbolic MoE outperforms various baselines.

**Compliance With Llm Reviewing Policy:**

Affirmed.

**Final Justification:**

the author response is nice

**Key Questions For Authors:**

please see above

**Limitations:**

yes

**Strengths And Weaknesses:**

Collaboration of multiple LLMs is an important research question. Some thoughts:

- The select-then-aggregate idea is classic, what the authors did is mostly proposing a new way of doing it: with performance profile-based selection, and dynamically choosing aggregators. It would be nice to have previous select-then-aggregate methods as baselines, some pointers include [1-2].

- I'm unsure about calling this "MoE": MoE seems to more often indicate MoE models: multiple experts, sparse activation, collaboration at the logit space, but still wrapped together as a single MoE model. Maybe there is a way to set the expectation straight at the beginning of the paper, that this "MoE" means a way for multiple models to collaborate?

- The authors posit that "Past work has often relied on multiple rounds of inference, leading to significant GPU demands. Moreover, it does not scale to a dynamic setting like the one we consider, where the number of GPUs required would be equal to the number of potential models available (in our case, 16), making this option prohibitively expensive." in line 100. *The number of GPUs required would be equal to the number of potential models available* is untrue. If users have fewer GPUs than the number of models, they can just round-robin assign models to GPUs in rotation, which is common practice.

- On that note, the authors seem to stress the "batch inference" thing as a novel way of serving multiple models. Actually, it seems quite standard practice (based on the figure 3), and the two settings on the left of figure 3, where batch inference supposedly outperforms, are only strawman baselines. Especially setting 2 in figure 3, who would do it like that?

- It would be nice to have at least one sentence describing each baseline in Section 4.1. Readers might feel disoriented and struggle to make sense of the many method names suddenly popping up in Table 1.

- Why couldn't we do multi-agent debate for "multi-model multi-agent", the last row group in Table 1? Just have each different model act  as a participant in the debate.

- Overall symbolic MoE seems like a clever way to re-engineer the select-then-aggregate pipeline with clever prompting: prompting to get the "skills" or problems (the accuracy of this step is not evaluated?), selection based on skills profile, again prompting an off-the-shelf model to aggregate, etc. Compared to things like [1-2] where some training was baked in the aggregation process, some might say the technical contribution is limited. The authors are free to argue otherwise.

[1] Jiang et al. "Llm-blender: Ensembling large language models with pairwise ranking and generative fusion."

[2] Zhao et al. "The majority is not always right: Rl training for solution aggregation."

---

> ### Author Rebuttal · Authors · 2026-03-30
>
> We sincerely thank the reviewer for their detailed and constructive feedback. Please find the response to each comment below.
>
> ---
> &nbsp;
> ### W1 & W7: Positioning w.r.t. prior Select-and-Aggregate approaches
>
> We agree that select-then-aggregate methods are important baselines and have clarified this in our evaluation. In fact, we already include closely related variants. In Table 7, we compare aggregation strategies under the same expert pool: **using a task-best model (a common select-then-aggregate baseline) is consistently weaker than our task-specific aggregator.** Table 4 further shows that simple majority vote over selected experts is a strong baseline, highlighting that **expert selection itself is critical, while combining it with a task-specific aggregator yields the best performance.**
>
> These results support two points:
>  (1) careful expert selection provides substantial gains over naive selection;
>  (2) aggregation further improves performance, but is complementary to selection.
>
> We thank the reviewers for the pointers. We have cited \[1\] and will add \[2\]. We attempted to include both as baselines, however:
>
> For LLM-Blender \[1\], the public GenFuser model is limited to 256 tokens for inputs and outputs and is trained mainly on instruction-following tasks. This causes severe truncation and domain mismatch in our setting, leading to near-zero performance.
>
> For AggLM \[2\], no public checkpoint is available. Reproducing it is infeasible due to the scale of training (128 rollouts over 40k samples, totaling 446,220 reasoning chains) and RL cost. Also, the aggregator used in our work is not specifically trained for aggregation, making this comparison less fair. That said, we'd like to highlight again that our method is complementary to the aggregator’s strength: stronger aggregators consistently improve results. Our contribution focuses on expert selection and is compatible with future aggregation advancement.
>
> ---
> &nbsp;
> ### W2: Clarification on the MoE terminology
>
> We agree that "MoE" is often associated with parameter-level mixtures. Our usage refers instead to (i) independently pre-trained models as experts and (ii) collaboration via a shared symbolic channel (language), rather than shared parameters.
>
> **We will add the following clarification:**
>
> “Clarification on terminology: We use ‘Mixture-of-Experts’ in a functional sense: selectively recruiting and combining specialized experts per input. Unlike conventional MoE, where experts are parameter subsets within a single model, our experts are independently pre-trained LLMs. They are not jointly trained and do not share parameters. Instead, we route at the model level based on inferred skills and aggregate via natural language. Our approach is thus a symbolic, training-free MoE framework.”
>
> ---
> &nbsp;
> ### W3 & W4: Clarification on the efficiency and novelty of the batched inference strategy
>
> **Efficiency.** While round-robin is practical, it does not address key challenges. It incurs heavy model loading/offloading for 7B–8B LLMs. In our adaptive setting (different experts *per instance*), this leads to frequent swaps and high latency.
>
> Our method precomputes assignments and groups instances by expert, so each model is loaded once per batch and runs a single batched forward pass. This removes repeated loading overhead and maximizes throughput on a single GPU, with straightforward scaling to more GPUs.
>
> **Novelty.** We clarify that the novelty is adaptive pre-routing, not batching itself. Standard batching assumes fixed model assignment; here, required models vary per instance via skill inference. This dynamic routing combined with grouped batching is, to our knowledge, not described in prior Multi-agent inference work.
>
> Practically, Setting II (Fig. 3\) is the default for many users who cannot host multiple 7B–8B models simultaneously and must process instances individually. Our approach (Setting III) provides a practical middle ground. **We will clarify that we do not claim to invent batching, but its integration with dynamic expert recruitment, and distinguish it from systems like vLLM that assume static model selection.**
>
> ---
> &nbsp;
> ### W5: Inclusion of baseline details
>
> Thank you for the suggestion! We will move baseline descriptions from Appendix C to the main paper in the final version.
>
> ---
> &nbsp;
> ### W6: Clarification on the multi-agent debate baseline
>
> The suggested multi-model debate setup corresponds to ReConcile \[1\], where different models act as agents producing answers with confidence, and the final prediction is determined via confidence-weighted consensus. This has been shown to outperform multi-agent debate with a single model, and Symbolic-MoE consistently outperforms it.
>
> \[1\] https://arxiv.org/abs/2309.13007

---

> > ### Author Rebuttal · Reviewer_tpjS · 2026-04-02
> >
> > I would like to thank the authors for the detailed response and adjust the score.

---

### Official Review · Reviewer_gQ5i · 2026-03-11

**Soundness:** 3
**Presentation:** 3
**Significance:** 2
**Originality:** 2
**Overall Recommendation:** 4
**Confidence:** 3

**Summary:**

The authors propose SYMBOLIC-MOE to select LLM-based expert adaptively at the instance level for heterogeneous reasoning tasks. Each expert is selected based on how relevant its expertise is to the query, and then generates its own reasoning. All experts' outputs will be synthesized into a final high-quality response by an aggregator, chosen based on its ability to integrate diverse outputs.

**Compliance With Llm Reviewing Policy:**

Affirmed.

**Final Justification:**

My main concerns were addressed.

**Key Questions For Authors:**

1) How does SYMBOLIC-MOE handle domains with scarce labeled data or where skill keywords are ambiguous?
2) Could dynamic updates to expert profiles during deployment improve performance?
3) Is there any observed degradation when input questions contain noisy, adversarial, or out-of-distribution phrasing?
4) Would using a universal aggregator instead of task-specific ones maintain performance?

**Limitations:**

yes

**Strengths And Weaknesses:**

Strengths
1) The approach for expert selection via skill expertise is well-defined and logically consistent.
2) Comprehensive evaluation on multiple benchmarks, including MMLU-Pro, GPQA, AIME, and MedMCQA
3) Tackles a key challenge in multi-LLM deployment, which is how to efficiently combine heterogeneous LLMs without retraining.

Weaknesses
1) The symbolic skill extraction relies on existing LLMs, which may not generalize to domains lacking structured skill definitions.
2) Dependency on validation sets and keyword LLM for accurate expert selection.
3) Limited exploration of failure modes, domain bias, or noisy input robustness.

---

> ### Author Rebuttal · Authors · 2026-03-30
>
> We thank the reviewer for recognizing that our approach is "**well-defined and logically consistent**", conducted "**comprehensive evaluation on multiple benchmarks**", and "**tackles a key challenge in multi-LLM deployment**".
>
> ---
> &nbsp;
> ### W1: Generalization of Symbolic-MoE
>
> We further evaluate on ARC-Challenge \[1\], a multi-step commonsense benchmark with a weaker skill structure:
>
> | Method                      | Acc.  |
> |-----------------------------|-------|
> | Task-Best Model             | 88.6  |
> | Self-Consistency (Best ×5)  | 89.3  |
> | Mixture-of-Agents           | 90.1  |
> | ReConcile                   | 89.0  |
> | Symbolic-MoE                | **91.7** |
>
> Despite reduced structure, Symbolic-MoE achieves the best performance, indicating that symbolic skills remain expressive for effective routing.
>
> [1] https://arxiv.org/abs/1803.05457
>
> ---
> &nbsp;
> ### W2 / Q1: Robustness to Limited Data and Ambiguous Skill Definitions
>
> We evaluate zero-shot transfer by reusing profiles from MMLU-Pro/AIME and testing on OmniMATH:
>
> | Model            | Profile from MMLU-Pro | Profile from AIME |
> |------------------|----------------------|-------------------|
> | Debate           | 34.51                | 42.93             |
> | MoA              | 31.55                | 47.36             |
> | Self-MoA         | 14.63                | 48.75             |
> | ReConcile        | 22.01                | 42.62             |
> | **Symbolic-MoE** | **49.32 (+14.81)**  | **52.03 (+3.28)** |
>
> With MMLU-Pro, baselines degrade under domain shift (broad → math), as they rely on task-level selection. Symbolic-MoE’s skill-level routing generalizes better, yielding a \+14.81 gain. With AIME, the gap narrows, but Symbolic-MoE still leads (+3.28), showing a consistent gain.
>
> **Robustness to Keyword LLM choice.**
> Our results show robustness to the choice of Keyword LLM. As shown in Table 15 (Lines 947–957), using Qwen 2.5 7B, Llama 3.1 8B, or Gemma 2 9B yields similar downstream performance.
>
> ---
> &nbsp;
> ### W3: Analysis of Failure Modes, Domain Bias, and Robustness
>
> **Failure modes.** We acknowledge that Symbolic-MoE may fail in several scenarios: (1) *Collective expert failure*, where none of the experts produces a correct answer, leaving the aggregator with little chance of recovery; and (2) *Aggregation errors*, where the aggregator selects an incorrect response. We provide statistics for these two kinds of errors on AIME and GPQA.
>
> | Dataset | Collective Expert Failure | Aggregation Error |
> |---------|---------------------------|-------------------|
> | AIME24  | 33.3%                     | 20.0%             |
> | GPQA    | 29.3%                     | 18.7%             |
>
> **Domain bias.** We evaluate across math (AIME), medical (MedMCQA), general (MMLU-Pro), STEM (GPQA), and commonsense (ARC). These results suggest that **Symbolic-MoE generalizes well across domains.**
>
> **Noisy input robustness.** Our aggregator outperforms majority voting (which proved to be more robust [2]), indicating robustness to noisy outputs.
>
> We will include these discussions in the final version.
>
> [2] https://arxiv.org/abs/2203.11171
>
> ---
> &nbsp;
> ### Q2: Profile updates are efficient and effective for performance gain
>
> Yes -- efficiently. Adding a new model only requires inference on a small validation set.
>
> To test this, we augment the model pool with a stronger model (Qwen3 14B) and compare performance with and without its inclusion on AIME and GPQA:
>
> | Setting     | AIME  | GPQA  |
> |-------------------|-------|-------|
> | w/o Qwen3 14B     | 68.88 | 57.78 |
> | w/ Qwen3 14B      | 72.27 | 59.15 |
>
> This shows seamless integration of stronger experts with consistent gains.
>
> ---
> &nbsp;
> ### Q3: Robustness to distribution shift and OOD inputs
>
> We evaluate OOD transfer (MMLU-Pro → OmniMATH):
>
> | Model         | Profile from MMLU-Pro        |
> |---------------|-----------------|
> | Debate        | 34.51           |
> | MoA           | 31.55           |
> | Self-MoA      | 14.63           |
> | ReConcile     | 22.01           |
> | Symbolic-MoE | **49.32 (+14.81)**|
>
> The results show that **Symbolic-MoE achieves the strongest OOD performance**, outperforming the second-best baseline by a large margin (**+14.81**), showing robustness under large distribution shifts.
>
> ---
> &nbsp;
> ### Q4: Aggregator is complementary to downstream performance
>
> Yes. Performance is largely preserved, but optimal results require task-specific aggregation.
>
> Table 7 shows that replacing the task-specific aggregator with a universal one (e.g., task-best) leads to only a moderate drop. Expert selection is the main driver, as switching from random to recruited experts yields large gains (31.82 → 51.52), while improving aggregation gives additional improvements (53.54 → 57.78).
>
> A universal aggregator can maintain much of the performance, but the best results are achieved when careful expert selection is paired with a task-specific aggregator.

---

> > ### Author Rebuttal · Reviewer_gQ5i · 2026-04-04
> >
> > I have no further questions.

---

> > > ### Author Response · Authors · 2026-04-04
> > >
> > > Dear Reviewer,
> > >
> > > Thank you very much for taking the time to read our rebuttal and for indicating that your concerns have been **fully resolved**. We also appreciate your note that you have **no further questions**.
> > >
> > > We noticed that your score has remained unchanged despite selecting “*Fully resolved.*” While we fully understand that any score adjustment is at your discretion, we wanted to gently check in in case this was unintentional or overlooked.
> > >
> > > We would greatly appreciate it if you could kindly revisit the score at your convenience.
> > >
> > > Thank you again for your time and thoughtful review.

---

### Official Review · Reviewer_3BcB · 2026-03-11

**Soundness:** 3
**Presentation:** 3
**Significance:** 4
**Originality:** 3
**Overall Recommendation:** 6
**Confidence:** 4

**Summary:**

Most multi-agent LLM frameworks select experts at a task level, but this paper addresses the issue that individual instances can require very different settings. The authors remedy this by introducing a routing scheme for each instance.

**Compliance With Llm Reviewing Policy:**

Affirmed.

**Final Justification:**

I think this is a very good paper

**Key Questions For Authors:**

See weaknesses

**Limitations:**

yes

**Strengths And Weaknesses:**

Strengths:

The method is well-motivated, and the experiments are clear.

Weaknesses:

The entire routing mechanism uses an LLM (Qwen2.5-7B) to produce "skill labels". I think it would be interesting to see what these skill labels actually look like in practice, and how noisy they are. Not necessarily a weakness, though.

This is a very short review, but I think the paper is quite good, and I would recommend acceptance

---

> ### Author Rebuttal · Authors · 2026-03-30
>
> We sincerely thank the reviewer for their positive feedback, recognizing our method "**is well-motivated**" and "**the experiments are clear**", and **recommending acceptance with score 5 with high confidence \= 4\.**
>
> ---
>
> We agree that understanding the nature and potential noise of the skill labels is important. We provide two complementary pieces of evidence:
>
> **(1) Robustness to the choice of Keyword LLM.**
>
> The framework is largely insensitive to which LLM is used to generate skill labels. Below, we show the accuracy on MMLU-pro and GPQA for three models, ranging from 7-9B.
> | Keyword LLM            | MMLU-Pro | GPQA  |
> |---------------|----------|-------|
> | Llama 3.1 8B | 64.19    | 56.62 |
> | Gemma 2 9B | 64.02    | 57.01 |
> | Qwen 2.5 7B  | 63.71    | 57.78 |
>
> Despite slight differences in annotation behavior across three different models with varying sizes, downstream performance remains stable. This suggests that the routing mechanism does not rely on highly precise or noise-free labels.
>
> **(2) Empirical structure of skill labels.**
>
> In practice, the extracted skills are interpretable and align well with human intuition:
>
> *AIME 2024 (Unique keywords: 75\)*
>
> | Skill            | Count | Percent |
> |------------------|-------|---------|
> | algebra          | 15    | 9.43%   |
> | number theory    | 13    | 8.18%   |
> | geometry         | 12    | 7.55%   |
> | combinatorics    | 9     | 5.66%   |
> | trigonometry     | 6     | 3.77%   |
>
>
> *MMLU-Pro (Unique keywords: 1053\)*
>
> | Skill          | Count | Percent |
> |----------------|-------|---------|
> | psychology     | 185   | 1.66%   |
> | finance        | 184   | 1.65%   |
> | economics      | 159   | 1.42%   |
> | thermodynamics | 134   | 1.20%   |
> | law            | 112   | 1.00%   |
>
> AIME is dominated by core math areas such as algebra, number theory, and geometry. MMLU-Pro exhibits a long-tail distribution over diverse domains (e.g., psychology, finance, law), reflecting its broad coverage. This indicates that the labels capture meaningful high-level structure rather than noise.
>
> We thank the reviewer for the suggestion to help consolidate our work further, and we will add this to the paper in the final version.

---

> > ### Author Rebuttal · Reviewer_3BcB · 2026-04-01
> >
> > Resolved - updated my score

---

### Official Review · Reviewer_rpv2 · 2026-03-12

**Soundness:** 2
**Presentation:** 2
**Significance:** 2
**Originality:** 2
**Overall Recommendation:** 4
**Confidence:** 3

**Summary:**

This paper introduces SYMBOLIC-MOE, a gradient-free, text-based Mixture-of-Experts framework that routes LLM experts based on fine-grained skill matching. It achieves notable accuracy gains while significantly reducing computational overhead via batched inference.

**Compliance With Llm Reviewing Policy:**

Affirmed.

**Final Justification:**

The authors have addressed my concerns.

**Key Questions For Authors:**

1.How does SYMBOLIC-MOE perform on unstructured reasoning tasks with implicit skill requirements, and what is its generalization to zero-shot skills not present in the validation data?
2.What are the performance and efficiency tradeoffs when scaling the expert pool beyond 16 models, or when using smaller LLMs for skill annotation and aggregation?
3.How does the framework’s accuracy degrade under noisy skill annotations, and are there any safeguards to mitigate this?

**Limitations:**

yes

**Strengths And Weaknesses:**

This work presents a novel, gradient-free approach to LLM expert orchestration, decoupling task routing from model fine-tuning and demonstrating strong performance-cost efficiency. The skill-based routing and batched inference design are well-motivated and empirically validated across diverse reasoning benchmarks. However, several concerns remain:

1.The framework relies on explicit skill annotation via a Keyword LLM, and its generalization to unstructured or open-ended reasoning tasks with ambiguous skill requirements is not fully explored.

2.The expert pool is limited to 16 open-source models, and the scalability to larger, more diverse model pools, e.g., proprietary or multimodal LLMs, is untested.

3.The robustness of skill inference to noisy annotations or low-resource Keyword LLMs is not analyzed, leaving uncertainty about real-world deployment reliability.

---

> ### Author Rebuttal · Authors · 2026-03-30
>
> We thank the reviewer for recognizing our work shows "**notable accuracy gains while significantly reducing computational overhead via batched inference**", "**demonstrating strong performance-cost efficiency**", "**well-motivated**" and "**empirically validated across diverse reasoning benchmarks**". We address the reviewer’s comments below.
>
> ---
> &nbsp;
> ### W1 & W3: Robustness and generalizability
>
> Our results show robustness to the choice of Keyword LLM. As shown in Table 15 (Lines 947–957), **using Qwen 2.5 7B, Llama 3.1 8B, or Gemma 2 9B yields similar downstream performance**, showing robustness and insensitivity to the choice of Keyword LLM.
>
> To evaluate generalization to less structured tasks, we include ARC-Challenge \[1\], a multi-step commonsense reasoning benchmark with *weaker domain structure* and *less separation of skills*:
>
> | Method                     | Acc. (%) |
> |----------------------------|----------|
> | Task-Best Model            | 88.6     |
> | Self-Consistency (Best ×5) | 89.3     |
> | Mixture-of-Agents          | 90.1     |
> | ReConcile                  | 89.0     |
> | Symbolic-MoE               | **91.7** |
>
> Even with reduced task heterogeneity, our method achieves the best performance, indicating effectiveness beyond structured settings.
>
> \[1\] https://arxiv.org/abs/1803.05457
>
> ---
> &nbsp;
> ### W2: Scalability of the model pool
>
> **Scaling is straightforward: adding a model only requires lightweight profiling** on a validation set to estimate its skill distribution (\~8 minutes on 300 samples with 4×A6000 GPUs). No further training is needed.
>
> We validate this by adding Qwen3 14B:
>
> | Model Setting     | AIME  | GPQA  |
> |-------------------|-------|-------|
> | w/o Qwen3 14B     | 68.88 | 57.78 |
> | w/ Qwen3 14B      | 72.27 | 59.15 |
>
> Performance improves consistently, showing seamless integration of stronger experts.
>
> Our model pool design follows three principles:
>
> - Excluding proprietary models avoids dominance that would obscure skill-based routing under meaningful complementarity.
> - Open-source models ensure accessibility and reproducibility.
> - We focus on unimodal LLMs, as benchmarks are text-only and do not require multimodal reasoning.
>
> Adding any new open-source models is easy and lightweight, as mentioned above.
>
> ---
> &nbsp;
> ### Q1: Generalization of Symbolic-MoE
>
> **Less structured tasks.** As discussed above, on ARC-Challenge, despite weaker skill separability, Symbolic-MoE achieves the best performance, showing that symbolic skill extraction remains effective.
>
> **Unseen tasks.** We reuse profiles from MMLU-Pro and AIME and evaluate on OmniMATH without re-profiling (Table 2, Lines 307–329):
>
> | Model            | Profile from MMLU-Pro | Profile from AIME |
> |------------------|----------------------|-------------------|
> | Debate           | 34.51                | 42.93             |
> | MoA              | 31.55                | 47.36             |
> | Self-MoA         | 14.63                | 48.75             |
> | ReConcile        | 22.01                | 42.62             |
> | **Symbolic-MoE** | **49.32 (+14.81)**  | **52.03 (+3.28)** |
>
> Symbolic-MoE transfers better, as skills generalize more robustly than task-level rankings.
>
> ---
> &nbsp;
> ### Q2: Performance and efficiency tradeoffs
>
> Adding a new model only incurs a one-time, fixed overhead for profile construction, which is lightweight (e.g., a few minutes on a small validation set as discussed above). Once profiled, the model can be seamlessly integrated into the pool without any further training.
>
> At inference time, efficiency is largely unaffected by the total pool size. The router recruits only a small subset of models (top-k) for each instance, meaning that the computational cost depends on k, not on the total number of available experts.
>
> Regarding smaller LLMs for skill annotation and aggregation, these components can be flexibly instantiated with lightweight models to further reduce overhead, as they only need to infer skills rather than full problem solving, and we do not observe a significant difference between a 7B, 8B, and 9B model when used as the Keyword LLM.
>
> ---
> &nbsp;
> ### Q3: Safeguards to mitigate noisy skill annotations
>
> Our framework demonstrates strong resilience against noisy skill annotations with different Keyword LLMs as discussed above, and we have two main safeguards within our methodology to mitigate noisy skill annotation: (1) We prompt the Keyword LLM to generate annotations for each question five separate times (lines 129-132). We then filter the results by retaining only those skills that appear more than once. (2) A model's profile is a cumulative dictionary aggregated over many instances; occasional noisy tags on individual questions would have a low chance of being sampled.

---

> > ### Author Rebuttal · Reviewer_rpv2 · 2026-04-03
> >
> > Thanks for your detailed rebuttal, I raise my score.

---

### Decision · Program_Chairs · 2026-04-30

**Decision:**

Accept (regular)

**Comment:**

Symbolic-MoE is a gradient-free Mixture-of-Experts framework that routes 16 pre-trained LLM experts using symbolic skill matching at the instance level. By inferring fine-grained skills from queries and selecting experts via skill-profile overlap, it synthesizes outputs through a task-specific aggregator. A batch inference strategy allows loading each model once, enabling 16-model deployment on a single GPU with competitive throughput. Evaluations on MMLU-Pro, GPQA, AIME, and MedMCQA show improvements over the best baseline. Zero-shot generalization results include OmniMATH and ARC-Challenge.

Reviewers identified several core strengths:
Instance-level routing is well-justified and overcomes task-level constraints.
The batch inference approach is a notable engineering success, enabling 16 models on one GPU.
Reliable gains were achieved on benchmarks like MMLU-Pro, GPQA, AIME, and MedMCQA.
A gradient-free design allows broad implementation without fine-tuning.

There is a legitmate concern regarding the novelty or substantialness noting the method re-engineers the select-then-aggregate pipeline via prompting. As batch inference follows standard practices and baselines like LLM-Blender and AggLM were not reproduced, the system's specific novelty remains undefined.